# Advanced Drug-Eluting Poly (Vinyl Chloride) Surfaces Deposited by Spin Coating

**DOI:** 10.3390/medicina55080421

**Published:** 2019-07-30

**Authors:** Oana Cristina Duta, Maxim Maximov, Roxana Trusca, Anton Ficai, Denisa Ficai, Cornelia-Ioana Ilie, Lia-Mara Ditu, Ecaterina Andronescu

**Affiliations:** 1Department of Science and Engineering of Oxide Materials and Nanomaterials, Politehnica University of Bucharest, 011061 Bucharest, Romania; 2SC Microsin SRL, 032364 Bucharest, Romania; 3Academy of Romanian Scientists, 050094 Bucharest, Romania; 4Department of Microbiology and Immunology, Faculty of Biology, University of Bucharest, 050095 Bucharest, Romania; 5Department of Microbiology and Immunology, Faculty of Biology, Research Institute of the University of Bucharest, 050107 Bucharest, Romania

**Keywords:** polyvinyl chloride, catheters, spin coating, surface modification, dicoumarol

## Abstract

*Background and objectives:* Medical devices such as catheters are used on a large scale to treat heart and cardiovascular diseases. Unfortunately, they present some important drawbacks (structure failure, calcifications, infections, thrombosis, etc.), with the main side effects occurring due to adhesion and proliferation of bacteria and living cells on the surface of the implanted devices. The aim of this work is to modify the surface of polyvinyl chloride (PVC), an affordable biocompatible material, in order to reduce these aforementioned side effects. *Materials and Methods:* The surface of PVC was modified by depositing a thin layer also of PVC that incorporates an active substance, dicoumarol (a well-known anticoagulant), by spin coating process. The modified surfaces were analyzed by Fourier-transform infrared (FT-IR) microscopy, Fourier-transform infrared (FT-IR) spectroscopy, Ultraviolet-visible spectroscopy (UV-VIS), and Scanning electron microscopy (SEM) in order to determine the surface morphology and behavior. The samples were tested for Gram-positive (*S. aureus* ATCC 25923) and Gram-negative (*P. aeruginosa* ATCC 27853) standard strains from American Type Culture Collection (ATCC). *Results:* The material obtained had a smooth surface with a uniform distribution of dicoumarol, which is released depending on the deposition parameters. The concentration of dicoumarol at the surface of the material and also the release rate is important for the applications for which the surface modification was designed. PVC modified using the proposed method showed a good ability to prevent salt deposition and decreased the protein adhesion, and the resistance to bacterial adherence was improved compared with standard PVC.

## 1. Introduction

Cardiovascular diseases are responsible for a large number of deaths and it is estimated that this number will grow in the coming years. Because of this, it is necessary to obtain some new or improved materials for preparing cardiovascular devices, including vascular grafts for bypass, stents, and heart valves that provide the desired interactions between the material and body fluids, especially blood. One of the most important classes of medical devices are catheters, which can be used as diagnostic tools and also as clinical tools [1]. Catheters are used in treating vascular diseases and can also provide fluid administration, parenteral nutrition, hemodynamic monitoring, blood sampling, and administration of drugs [2,3,4,5,6,7,8,9]. Also, such devices have some important drawbacks, such as structure failure, calcifications, infections, thrombosis, etc. [2,3,4,5,6,7,8,9,10]. Thrombogenicity due to incompatibility of the material surface and the change in the flow dynamics at the site of implant are the most common causes that lead to failure of the device [11,12]. Intra-cardiac catheter introduction for experimental or diagnostic purposes and its maintenance for a period of 7–14 days can cause valvular or parietal endocarditis [13,14]. To overcome these drawbacks, many studies have been conducted on several materials. Among the most important polymers used as biomaterials in the medical field are polyurethane, silicone rubber, ethylene vinyl acetate, polyvinyl chloride (PVC), polycarbonate, polyester, polyacrylonitrile, hydrogels, etc. [7,15,16,17]. One of the most-used materials is PVC, due to its unique properties (mechanical strength and chemical resistance, is inert to biological fluids, has anti-aging properties, presents a wide range of possibilities for processing (mixing, pouring, extrusion, etc.), and is reasonably priced, even for the material devoted to medical applications [5,7,16,18].

A technique used to modify the surface of the material in order to obtain materials with improved properties so they can be used in biomedical applications is to cover the material with a polymeric thin film. This film should be uniform and should ensure a smooth non-thrombogenic surface with antibacterial properties. One of the methods used for this purpose is the method called “spin coating”. Spin coating is a physical process used to create uniform thin films (at the micrometric or nanometric level) on a flat surface (ceramic, metallic, or polymer surfaces) using centrifugal force [19,20,21,22,23]. The biggest impacts on the final thickness of the coating are the flow control stage (removal of excess solution) and evaporation [20]. The film-forming process is mainly driven by two independent parameters: the viscosity and the speed of rotation [20]. The biggest advantage of using the spin coating process is the absence of coupled process variables (variables that depend on one another).

Over the years, many studies have been performed to modify the surface of several materials using spin coating in order to improve their antibacterial properties. Zodrow et al. [24] the used spin coating method to prepare poly (lactic-co-glycolic) acid films with incorporated cinnamaldehyde (CA) and carvacrol (CARV) (natural antimicrobial compounds). They concluded that both CA and CARV could be used in polymer coatings to reduce biofilm formation by *Escherichia coli* and *S. aureus*, but not in the case of *Pseudomonas aeruginosa*. Elaheh et al. [25] prepared coumarin polyesters using spin coating and demonstrated that the cationic coumarin polyester could prevent biofilm formation without presenting toxicity towards mammalian cells. Nanoparticles generated great interest in this field because they have a high surface-to-volume ratio, thus enhancing their antimicrobial activity [26]. In time it was proven that silver nanoparticles significantly reduce bacterial adhesion on medical devices and are used in implants for surface modification [27,28,29,30]. Nevertheless, silver nanoparticles can produce dose-dependent cytotoxicity [27,29,30]. Spin coating technique was also used to prepare coatings that incorporate various types of nanoparticles. Among the nanoparticles studied we can mention nanoparticles of hydroxyapatite coatings modified with silver or zinc ions onto the Ti-6Al-4V substrate in order to improve the antibacterial activity and biocompatibility [31]. Another sol-gel coating using nanoparticles that exhibited reduced bacterial adhesion was silver-doped phenyltriethoxysilane [32]. Antimicrobial hydroxyapatite nanoparticles and chitosan composites were spin coated on 316 L stainless steel implant and showed improved antimicrobial resistance and biocompatibility [33].

The aim of this work was to modify the surface of polyvinyl chloride (PVC) so that the resulting material would not allow cell adherence and proliferation of bacteria on its surface and to prevent blood clotting or calcification. This is very important when the material is designed to be used in the medical field to obtain medical devices such as catheters. The most common problems encountered in the case of catheters are due to blockages or an infection at the site of implantation. Because of this, catheters must be removed and replaced frequently. This requires surgery, which involves high costs and causes serious discomfort to the patient. To achieve the desired objective, the surface of the PVC was physically modified by the method of spin coating in order to remove the unevenness arising from the manufacture of the PVC and to obtain a smooth, uniform surface. The smoother the surface, the more unlikely that cellular organisms will adhere to the material surface. If the material has unevenness, bacteria or cells can anchor around them because they cannot be washed by biological fluids, as the polymers are hydrophobic. In order to prevent blood clotting, the polymeric film deposited by spin coating contains an active substance from the class of coumarinic compounds, dicoumarol. This class of substances not only has anticoagulant properties, but also has antibacterial, antioxidant, and anti-allergic activities [34,35]. 

## 2. Materials and Methods

Here, 4-Hydroxycoumarin (Merck), formaldehyde (Chemical Company), ethanol (Chemical Company), piperidine (Merck), bovine serum albumin (Fluka), and standard flat samples of PVC and tetrahydrofuran were used without further purification. 

### 2.1. Dicoumarol Synthesis

Dicoumarol used in this work was synthesized in the laboratory. For the synthesis of dicoumarol and its derivatives, several synthetic routes are described in the literature [36,37,38,39]. In this case, the method used was the one proposed by Khan et al. [36] and consists of reacting 4-hydroxycoumarin at room temperature using ethanol as the reaction medium and catalytic amounts of piperidine (Figure 1). 

### 2.2. PVC/Dicoumarol Film Deposition

The PVC/dicoumarol solution to be deposited on the surface of a flat sample of PVC was prepared by dissolving 0.1 g dicoumarol (synthetized and purified in the laboratory) in 25 mL of tetrahydrofuran (THF) under magnetic stirring at 52 °C. After complete dissolution of the solid in the clear solution, 2.0 g PVC were added (from the same batch), cut into small pieces, and stirring continued until complete dissolution of the PVC. Thus, the obtained solution was dosed in small portions (~0.2 mL/dose) to cover the flat samples of 2 cm × 2 cm PVC using the spin coater apparatus. In order to obtain the optimum conditions for surface coating of the PVC samples, the parameters, such as the spinning time, rotation speed, and acceleration speed, were varied. Table 1 presents the values for which the results were most satisfying.

The samples were placed on the stage and vacuum was applied. The solution to be deposited on the surface of the sample was loaded into a syringe, which was then placed in the lid of the device. With the help of the syringe, the desired amount of solution was dosed (~0.2 mL/dose) and then the spinning process was started. After all of the steps of the spin coating process were performed, the vacuum was released and the sample was carefully removed and allowed to dry at room temperature. Thus, the samples obtained were analyzed by FT-IR spectroscopy in attenuated total reflectance (ATR) mode in order to confirm the presence of dicoumarol in the obtained films. The surface morphology was investigated using FT-IR microscopy to observe the changes that took place after the coating was performed, and also to see if the distribution of dicoumarol on the surface was uniform.

The pH of human blood is between 7.35 and 7.45 and is maintained in this range by keeping the balance between bicarbonate ions and carbonic anhydrase [40,41,42]. The normal temperature of the blood is about 36.5 °C [42]. Therefore, in order to mimic these conditions and to observe the surface behavior when in contact with blood and biological fluids, a solution of simulated body fluid (SBF) at pH 7.4 was prepared by the method described by Oyane et al. [43] and the in vitro tests were performed at 37 °C. 

The samples were characterized using a Thermo Scientific Nicolet iN10 Infrared Microscope in order to analyze the surface of the samples, and a Thermo Fischer Nicolet iN10 Attenuated Total Reflection Infrared spectroscope to determine the spectra of the obtained samples and to see if the surface composition had changed. The Ultraviolet-Visible spectrophotometry was performed on a Thermo Scientific Evolution 300 Spectrophotometer in order to determine the dicoumarol release from the surface in simulated body fluid—SBF solution. Scanning Electron Microscopy (SEM) and The Energy Dispersive Spectroscopy (EDS) analysis were performed in order to have a better view of the salts deposited on the surface. 

Protein adsorption on the surface was studied using FT-IR spectroscopy and microscopy after several periods of immersion in an SBF solution loaded with albumin, thus mimicking the conditions in the human body. 

The anti-adherent capacity of the samples obtained in this work was performed by determining the (colony-forming unit) CFU/mL values. The samples were tested for two types of standard strains: Gram-positive (*Staphylococcus aureus* ATCC 25923) and Gram-negative (*Pseudomonas aeruginosa* ATCC 27853). In the performed tests, the 18–24 h fresh cultures were obtained after inoculation of the bacterial strains on Nutrient Agar medium. The PVC samples were numbered in order to ease the marking (Table 2). 

#### 2.2.1. Experimental Procedure

The samples were cut to approximately equal sizes (1 cm/1 cm) and were pre-sterilized by UV radiation (30 min on each side of the material) in order to avoid the impact of contaminants on the experiment. Subsequently, the samples were placed in Petri plates on the surface of a medium of 2% Agarified Gelose (for bacterial species), with a thickness of 4 mm, with the untreated part fixed to the surface of the medium and with the treated side exposed on the outside. Microbial cell suspensions in sterile physiological water were prepared from fresh cultures, with a standard density of 1.5 × 10^8^ CFU/mL (corresponding with 0.5 McFarland standard). To treat the samples, microbial cell suspensions were prepared in nutrient broth medium in order to obtain a standard density of 1.5 × 10^7^ CFU/mL. Subsequently, 100 μL of cellular suspension was added dropwise over every sample and the incubation was performed in humid atmosphere at 37 °C for 24 h. During this time the microbial cells initially multiplied in the solution, and after reaching a density threshold they started to adhere to the material surface.

#### 2.2.2. Quantitative Assessment of the Selected Strains Capacity to Adhere to the PVC Surfaces

Quantitative assessment of the capacity of selected strains to adhere to the material surface was performed using decimal microdilution method. After 24 h, each sample was extracted from the medium, gently washed dropwise with 1 mL of sterile physiological water in order to remove non-adherent cells, and then placed in sterile centrifuge tubes containing 1 mL of sterile physiological water. Samples were vortexed for 30 s for mechanical detachment of adherent bacterial cells from the materials surfaces [44], followed by the decimal dilutions of the resulting suspensions. In order to determinate the CFU/mL (colony forming units/mL) values, each dilutions was inoculated, in triplicate, on the surface of 2% agar medium incubated for 24 h at 37 °C. The CFU/ml values were expressed as the average of the total number of colonies × 1/D (D = decimal dilution, for which the number of total colonies was determinate) [45].

## 3. Results

In order to confirm the presence of dicoumarol on the surface of the modified samples, the FT-IR spectra of the obtained surfaces were recorded and were compared with the spectrum of simple PVC, and respectively of the pure dicoumarol. Figure 2 presents the spectra of the samples obtained at drying speeds of 4000 rpm, 7000 rpm, and 10,000 rpm, respectively. 

In the obtained spectra the appearance of the characteristic peaks of dicoumarol at 1650 cm^−1^ and 1350 cm^−1^ can be observed, corresponding to ν(C=O) and ν(C–O) bounds of the lactone rings, respectively. Also, the peaks present at 1110–1130 cm^−1^ indicate the presence of the ν(C–OH) bound and at 1070 cm^−1^ the ν(C–O–C) bound. 

In Figure 3a–c, the images obtained by FT-IR microscopy of the samples obtained at 4000, 7000, and 10,000 rpm, respectively, are presented. FT-IR analysis was performed on the plates subjected to spin coating deposition process at the limit of the flow of the coating solution, meaning to the extent at which the coating was performed, in order to easily highlight the morphological change of the initial uncoated surface.

The video image of the sample modified by spin coating at 4000 rpm (Figure 3a) reveals a flawed surface (“hill-valley” appearance), which is corrected by deposition of the film. Basically, this surface no longer has the appearance of a “hill-valley”, as all defects were covered by the polymer solution. The FT-IR image recorded at 1655 cm^−1^, the characteristic wavelength of dicoumarol, reveals a relatively uniform distribution of the active substance on the surface of the PVC film. The video image recorded for the sample modified by deposition at 7000 rpm (Figure 3b) was also analyzed at the flow limit of the coating solution so it is possible to see two totally different morphologies. The initial substrate, which has not been modified with PVC/dicoumarol solution, presents spherical defects up to 150–250 μm in diameter, but after the process of spin coating was performed at 7000 rpm, these defects virtually disappeared. FT-IR images recorded at 1650 cm^−1^ reveal a good distribution of dicoumarol in the deposited film. In the case of the sample obtained at 10,000 rpm, the defects were completely covered by the polymer solution. The FT-IR image (Figure 3c) shows the best homogeneity of dicoumarol in the deposited layer, even at the limit of flow.

### 3.1. Release of Dicoumarol from the Thin Film Deposited on the Surface of PVC

In order to observe if the drug was released from the film, each sample was immersed in 100 ml of SBF solution at 36.5 °C and maintained for 30 days. The SBF solution was analyzed by UV-VIS at 303 nm, the wavelength for which the absorbance of dicoumarol is the highest, at different periods of time (1 h, 2 h, 4 h, 24 h, 2 days, 3 days, 7 days, 10 days, 14 days, 21 days, and 30 days) to observe the excite behavior of dicoumarol in the PVC film. In order to have comparable results, the samples were also analyzed by FT-IR microscopy to determine if the surface morphology changed over time when in contact with the SBF solution. FT-IR microscopy was also used to confirm the presence or absence of dicoumarol on the surface after several days. The UV-VIS results were represented in the release chart corresponding to the samples obtained at 4000 rpm, 7000 rpm, and 10,000 rpm, respectively (Figure 4). 

According to the UV-VIS analysis, the release of the active principle takes place in a similar way for all three samples. The release takes place rapidly from the surface layer in the first hours and then dicoumarol is released gradually from the PVC film in the next days as a new amount of anticoagulant is exposed due to material swelling. The samples obtained at 4000 rpm and at 7000 rpm present a more rapid release of dicoumarol from the surface than the sample obtained at 10,000 rpm in the first 2 days. In the next days, the release occurs gradually from all three samples, however, the UV-VIS analysis showed a more uniform and more gradual release in time for the sample obtained at 10,000 rpm. In the first 10 days, a larger quantity of dicoumarol is released from the sample obtained at 7000 rpm, but then the release takes place in the same manner as the other samples. 

The FT-IR microscopy allows one to observe the morphology of the surface and the local spectrum in any of the analyzed points of the surface. According to this, the dicoumarol presence and distribution on the surface of the modified material can be evaluated. In time, even if the morphology of the surfaces of the samples does not change, a significant decrease of intensity of the corresponding peak of dicoumarol is observed, and also the appearance of deposits on the surface, which become more obvious over time, eventually covering the whole surface. This evolution can be observed in Figure 5 for the film obtained at 4000 rpm, in Figure 6 for the film obtained at 7000 rpm, and in Figure 7 for the film obtained at 10,000 rpm. The deposition takes place more rapidly on the surface of the sample at 7000 rpm than on the sample at 4000, but in the case of the sample at 10,000, even after 30 days the deposits are slightly noticeable. The simple PVC sample showed similar behavior to the samples at 4000 and 7000 rpm, so the film deposition did not reduce salts deposits. According to the FT-IR spectrum performed after 30 days of immersion in SBF, dicoumarol is still present on the surface of the sample at 10,000 rpm, which could mean that its presence influences salt adhesion on the surface. 

The drug release takes place over a longer period of time in the case of the sample obtained at 10,000 rpm and in a more constant way, meaning that this method of altering the surface of PVC can be successfully used to incorporate dicoumarol on the surface of the material. Dicoumarol was chosen because of its anticoagulant and antibacterial activity, in order to obtain a material with improved surface properties, which can be used for medical devices that come in contact with blood and body fluids, such as catheters. From the FT-IR microscopy images, we can conclude that in the case of the sample obtained at 10,000 rpm, the salt deposition was significantly decreased. This aspect is important because these depositions can lead to the formation of calcifications resulting in catheter blocking, or to platelet activation responses. 

It is important to determine if the formulation of the material allows the adhesion of proteins on the surface or influences the platelet response. As in the previous literature, in the first days there were no significant changes on the surface of the thin films, but after 7 days of immersion in SBF, the surface appearance started to change, gaining a slightly heterogeneous appearance due to the deposition of salts and becoming less uniform, as can be seen in Figure 5a, Figure 6a, and Figure 7a, corresponding to samples 4000 rpm, 7000 rpm and, 10,000 rpm, respectively. In the case of the sample obtained at 7000 rpm, after 7 days of immersion in SBF, some deposits can be observed on the surface (Figure 6a), while for the samples obtained at 4000 rpm (Figure 5a) and at 10,000 rpm (Figure 7a), the deposits are almost unnoticeable. After 21 days, the deposition is very obvious also on the sample obtained at 4000 rpm (Figure 5b). The sample obtained at 7000 rpm is almost completely covered after 21 days (Figure 6b), unlike the sample obtained at 10,000 rpm, on whose surface the deposits are slightly noticeable (Figure 7b). 

After 30 days of immersion in SBF solution, the samples obtained at 4000 rpm (Figure 5c) and at 7000 rpm (Figure 6c) are completely covered with white deposits, and because of this they have a rougher surface. The sample obtained at 10,000 rpm behaves similarly to the other samples in the first days, but after 7days the differences are very obvious. This sample, even after 30 days of immersion in SBF, presents a small quantity of depositions on the surface (Figure 7c) and the surface smoothness does not change as much as in the case of the other samples. The FT-IR spectra show the presence of dicoumarol on the surface and the microscopy image reveals a uniform dispersion of the drug, indicating a uniform release from the polymer film. 

In order to see if the changes performed on the material influenced the salts’ adhesion on the surface, a sample of simple PVC was also immersed in SBF for 30 days and was analyzed by FT-IR microscopy at 1723 cm^−1^ (the characteristic wave length of the simple PVC) at the same periods of time as the modified samples. Simple PVC samples started to present deposits on the surface after 7 days (Figure 8a), and after 21 days (Figure 8b) and 30 days (Figure 8c), the surface is almost completely covered. Comparing the surface of the simple PVC after immersion in the SBF solution for several periods of time with the surface of the modified samples, we can observe that between this sample and the samples obtained at 4000 rpm and 7000 rpm, there are no significant differences, so the surface modification does not improve the surface properties regarding salt deposition. However, in the case of the sample obtained at 10,000 rpm, the differences are major, with the deposition being very reduced, even after 30 days. Scanning Electron Microscopy (SEM) and The Energy Dispersive Spectroscopy (EDS) analysis were performed in order to have a better view of the salts deposited on the surface (Figure 9) and the composition of these deposits. Figure 10 presents the spectrum obtained by EDS analysis and it can be seen that the deposits contain calcium, chloride, oxygen, sodium, and phosphorus, so most likely the salts present on the surface are made of sodium and calcium chloride, phosphates, carbonates, or oxides. 

The FT-IR spectra obtained with the FT-IR microscope does not give clear information about the presence of dicoumarol in the film, thus, the surface was analyzed also by FT-IR spectroscopy in ATR mode after different periods of time (Figure 11).

From the FT-IR spectrum of the sample obtained at 4000 rpm, it can be observed that even after 3 days of immersion in SBF, the characteristic peak of dicoumarol (~1650 cm^−1^) can be highlighted. Even so, we can see from the release chart obtained after the UV-VIS analysis that the polymer gradually allows the release of the drug over time as the swelling takes place, meaning that dicoumarol concentration at the surface is very low but the delivery continues from the bulk. In the case of the sample at 4000 rpm, the polymeric film is thicker, so the drug release requires more time than the other samples. In the case of the sample obtained at 7000 rpm, after 14 days the characteristic peak of dicoumarol at around 1654 cm^−1^ is still present, indicating that a quantity of drug remained on the surface, but after 30 days, as in the case of the sample obtained at 4000 rpm, the concentration of the dicoumarol from the surface is low, but it is possible that a quantity of dicoumarol is still present inside the film, as the concentration tends to grow in the SBF solution in which the sample is immersed, as indicated by UV-Vis study (Figure 4). Unlike these two samples, the sample obtained at 10,000 rpm contains dicoumarol incorporated on the surface even after 30 days. This film is thinner and has a more uniform distribution of the drug, so the release takes place more evenly because the drug release takes place progressively. The relative area of the peak corresponding to dicoumarol was calculated as the ratio between the area of the corresponding peak of dicoumarol (1650cm^−1^) and the area of the corresponding peak of PVC (1720 cm^−1^), and then multiplied by 100. This area was calculated in order to estimate the drug release trend from the deposited polymer film. From Figure 12, it can be observed that the release from the surface takes place rapidly, but in time, the polymer slowly expands and exposes a new quantity of dicoumarol on the surface. In the case of the sample obtained at 4000 rpm, the release takes place from the first days, so that after 3 days the drug was completely released from the PVC film. The samples obtained at 7000 rpm and at 10,000 rpm after 14 days still contain dicoumarol on the surface, but after 30 days only the sample obtained at 10,000 rpm has the active principle incorporated due to the more compact, denser structure of the thin film deposited. 

### 3.2. Protein Adsorption on the Surface of PVC

A very important aspect when using a material for medical devices, such as catheters, represents the way it reacts with the components of body fluids, especially with the salts and proteins. In order to study the behavior of the material obtained by coating with a thin film of PVC/dicoumarol during the interaction with proteins in the body fluids, the samples were immersed in a solution of SBF with a pH value of 7.4 containing 1% albumin (the main protein encountered in blood) at 36.5 °C and maintained at this temperature for 30 days. During this time, the samples were analyzed by FT-IR spectroscopy on the surface to observe if the sample morphology underwent changes over time and to see if the characteristic peaks of albumin appeared, meaning the deposition took place. A sample of PVC covered with albumin was prepared by the drop cast method in order to highlight the characteristic peaks of albumin and to see if they appeared on the spectra of the samples immersed in SBF/Albumin solution. Therefore, in Figure 13 the comparison between the spectra of the samples after 7 days of immersion, the initial uncovered PVC, and the sample obtained by the drop cast method is presented.

The most obvious different peaks present in the PVC/Albumin surface are the peaks at 1540–1600 cm^−1^ and around 1650 cm^−1^, with the latter being also present in the dicoumarol spectrum, so the chosen peak was the one at 1540–1600 cm^−1^. After these first 7 days, all of the samples showed a small amount of albumin deposited on the surface, but the peak area of the characteristic peak of albumin in the case of the uncovered PVC was larger compared to the modified samples. The sample at 4000 rpm had the lowest degree of deposition after 7 days according to the FT-IR spectrum, while the samples at 7000 rpm and 10,000 rpm were very similar. To see if the protein adhesion continued and in what degree, the samples were analyzed after another 2 and 3 weeks, however, the tendency remained the same. 

To have a more clear view of the way the material reacts when in contact with the protein solution, and to determine if albumin is adsorbed on the surface of the material and resulting in changes of morphology, the samples were also analyzed by FT-IR microscopy (Figure 14). It can be observed that after exposure in albumin solution for longer periods of time, the uncoated PVC is uniformly covered on the surface and starts to create a protein layer; the other samples also present an albumin layer, but the distribution is not so evenly distributed and the thickness of the protein film increases more slowly.

### 3.3. Bacterial Adhesion

In this study, the samples were examined against *S. aureus* and *P. aeruginosa*. As expected, in the case of gram-positive strain *S. aureus* (Figure 15), for the samples that have dicoumarol incorporated on the surface (1, 3, 5), a decrease in CFU/mL values may be observed with at least 4 logarithmic units. Also, a decrease in CFU/mL values with 3 logarithmic units may be observed in the case of sample 6, which represents the sample obtained at 10,000 rpm. This was the sample with the slowest degree of drug release, which even after 30 days immersion in SBF/Albumin solution had dicoumarol present on the surface and had the best capacity to inhibit protein adhesion from the three analyzed samples. 

In the case of *Pseudomonas aeruginosa* ATCC 27853 strain, an increase of all CFU/mL values can be observed (Figure 16), thus the Gram-negative strain adhered to the surface of all analyzed samples. 

## 4. Discussions 

Medical devices have some important drawbacks, such as structure failure, calcifications, infections, thrombosis, etc. [2,3,4,5,6,7,8,9,10], which are most often due to incompatibility of the material surface. Polymers are widely used as biomaterials in the medical field [7,15,16,17]. In this work, PVC was chosen because in the previous literature it was described as a material with good biocompatibility that would present good properties for the intended purpose [5,7,16,18] and also is reasonably priced. Although it is a material that is suitable for medical applications due to its properties, the commercially available PVC has a rough surface that can promote bacterial adhesion. In order to correct these surface defects, in this study the spin coating technique was used with the aim of obtaining a smooth, uniform surface without imperfections where proteins and bacteria can bind. Also, dicoumarol was used in the formulation of the solution to be deposited in order to improve antibacterial properties of the surface, and at the same time to prevent blood clotting and thrombus formation. 

After the spin coating was performed, the samples were tested to determine the surface morphology and to determine the presence and distribution of dicoumarol in the deposited film. As expected, the FT-IR spectroscopy (Figure 2) and FT-IR microscopy (Figure 3) confirmed the presence of dicoumarol with a good distribution on the surface. The spin coating process has four main stages: fluid release, spin-up, fluid outflow, and evaporation, steps which are described in detail in several literature papers [19,20,21,22]. The biggest impacts on the final thickness of the coating are the flow control stage (removal of excess solution) and evaporation. Dicoumarol distribution on the surface was influenced by these parameters. Homogeneity in the deposited layer increased with increase in the evaporation speed; the sample at 10,000 rpm showed the best homogeneity of dicoumarol in the deposited layer, even at the limit of flow. The video image of the uncovered PVC plates revealed a surface with a hill-valley appearance but was corrected by the film deposited by spin coating. 

It is important to determine if the formulation of the material allows the adhesion of proteins on the surface or influences the platelet response. According to previous literature, dicoumarol shows a greater degree of binding with albumin due to the presence of the hydroxyl group of 4-hydroxycoumarin that generates a negatively charged oxygen atom that can undergo an electrostatic interaction with the cationic centers of albumin [46]. It can be observed that in the case of the samples prepared by spin coating, the albumin adheres to the surface when dicoumarol is exposed at the surface of the thin film, confirming previous works in the field, but is washed when the drug dissolution takes place. 

The bacterial infections of implanted devices generally occur in several steps, starting with the bacterial adhesion to the material surface [47]. Bacterial adhesion and proliferation are influenced by various factors, such as environmental factors (pH, temperature, bacterial concentration, nutrient availability, flow conditions), the presence of serum proteins or antibiotics, the bacterial properties, and the characteristics of the material surface (surface charge, hydrophobicity, surface roughness, physical configuration) [48,49]. A very important factor that promotes bacterial attachment is the surface roughness, because the contact area between the material surface and bacteria cells increases due to irregularities, and also these surface flaws provide a protective shelter for colonization. Thus, the biofilm formation can be reduced by smoothing the surface, but other environmental factors must also be considered [48,49,50]. Bacterial adhesion is also strain dependent and tissue cell adhesion is substratum dependent (4). Surface hydrophilicity is very important, as it was observed that the interaction of tissue cells with hydrophobic polymers is greater than with hydrophilic polymers [47,51,52]. 

In natural conditions, proteins and bacteria are most often concomitantly present in complex biological fluids. Studies performed in the biological domain have revealed that protein coated surfaces may reduce bacterial adhesion or may provide a better fixation of biofilms [48]. As an example, Bovine Serum Albumin interacts non-specifically with bacteria, thus pre-coating the surface with albumin leads to a decrease of bacterial retention [48,51,53], while Fibronectin and Fibrinogen lead to an increase of bacterial retention [48,52]. Protein adsorption to the surface can occur through bio-specific or non-specific interactions, such as Van der Waals, Lewis acid/base, and electrostatic interactions [48,54]. 

The most encountered nosocomial infections are associated with *Escherichia coli* and *Staphylococcus aureus* [54], but there are many other gram-negative or gram-positive strains that must be taken into consideration. Infections related to implanted cardiovascular devices are mostly caused by gram-positive bacteria, such as staphylococci [47]. *S. aureus* is a gram-positive strain that can cause skin infections and respiratory diseases. *P. aeruginosa* is a gram-negative strain that can cause bronchial tract infection or diseases such as cystic fibrosis [55]. PVC is a biomedical polymer that is prone to adherence and proliferation of various bacteria strains because it is a hydrophobic material with moderate interfacial tension [52], which can lead to serious nosocomial contaminations. Surface modification is a very studied method of improving the anti-adherent properties of PVC [56]. Herrero et al. [54] studied the behavior of bacterial adhesion to PVC films and discovered that the hydrophobic character of both the PVC and bacteria *S. aureus* results in a good coverage with this strain. *S. aureus* infection produces platelet and leucocyte coverage of the same nature [57]. 

Over the years, studies have been performed on coumarin-based compounds, including dicoumarol, in order to determine their effect on bacterial deposition. In blood, a drug may be bound to proteins (e.g., albumin) or unbound, influencing the drug’s effectiveness. Binding of serum albumin to coumarinic compounds is weak and is due only to the hydrophobic interaction, however dicoumarol shows a greater degree of binding with albumin due to the presence of the hydroxyl group of 4-hydroxycoumarin that generates a negatively charged oxygen atom that can undergo an electrostatic interaction with the cationic centers of albumin [46]. Researchers in the field studied the effect of dicoumarol on bacterial deposition using both gram-positive and gram-negative strains and concluded that this compound exhibits a high activity against cell growth but only against gram-positive bacteria, such as *S. aureus*, *Bacillus anthracis* and *Streptococcus pyogenes* [57,58]. In this work, the resistance of the modified surfaces against *S. aureus* and *P. aeruginosa* was studied and concluded that all samples that were exposed to protein adhesion had no inhibitory response for *S. aureus* strain, meaning that the presence of albumin enhanced the adhesion of bacteria to the surface. In the case of *P. aeruginosa*, none of the samples presented inhibitory activity. Similarly, Rehman et al. [59] reported moderate activities against *Pseudomonas aeruginosa* and other Gram negative bacteria, such as *Escherichia coli*, in their study regarding the different coumarin derivatives.

The sample obtained at 10,000 rpm was the sample with the slowest degree of drug release, which even after 30 days immersion in SBF/Albumin solution had dicoumarol present on the surface and had the best capacity to inhibit protein adhesion of the three analyzed samples.

## 5. Conclusions

PVC surfaces can be successfully modified by spin coating using various spin rates. The samples were analyzed on the coated surface and also on the uncoated surface. Based on the FT-IR microscopy it can be concluded that the surface morphology is significantly changed when different rotation speeds (4000, 7000 and respectively 10,000 rpm) are applied. The initial surface was rough, uneven, and had “hill-valley”-type defects that were covered by the dicoumarol-loaded polymeric film, resulting in a smooth, uniform surface. The changes are similar in all three cases, but the smoothest and most uniform surface was obtained at 10,000 rpm. 

After comparing the results of the analysis performed on all three samples, it can be concluded that the best results were noticed on the sample obtained at 10,000 rpm in terms of surface morphology, as dicoumarol distribution on the surface was better and the release was slower and more constant than in the case of the samples obtained at 4000 rpm and 7000 rpm, respectively. This sample showed a good ability to prevent the deposition of salts on the surface, while the other two samples behaved similarly to the uncoated material. Albumin adhesion takes place on all the studied samples, but the presence of dicoumarol and the smoothness of the surface obtained after the deposition decreased the protein adhesion onto the surface of the modified PVC. It is important to mention that the proposed methodology induced a better anti-adherent ability of the PVC surfaces, especially against Gram positive strains, such as *S. aureus*, and the resistance to bacterial adherence was improved by incorporating dicoumarol on the surface due to its antibacterial properties and protein binding capacity. 

The spin coating method cannot be applied on tubular devices, such as catheters, but some principles used in coating techniques can be applied using an alternative method that can mimic the deposition conditions studied in the spin coating method and that can be used on tubular devices. The study of such a method is an objective for our future studies and the purpose is to develop a technique that consists of circulating a solution (prepared as described in this paper) through a tubular device, similar to a catheter, with a certain flow, using a peristaltic pump, so that the results obtained in this work can also be reproduced in the case of tubular devices, such as catheters. 

## Figures and Tables

**Figure 1 medicina-55-00421-f001:**
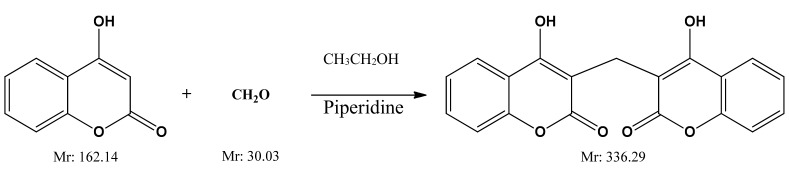
Dicoumarol synthesis route.

**Figure 2 medicina-55-00421-f002:**
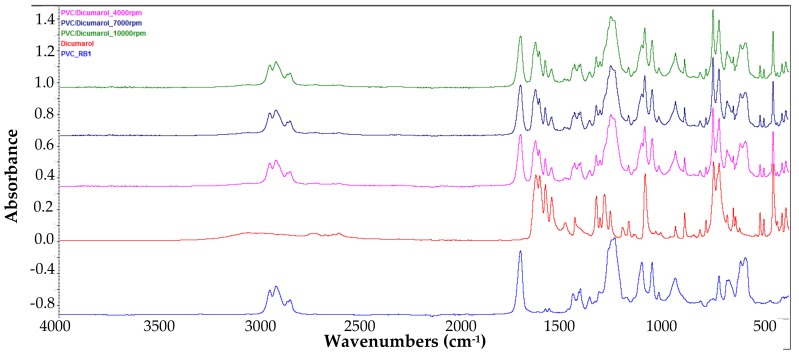
FT-IR spectra of the film obtained at 4000 rpm, 7000 rpm, and 10,000 rpm, respectively, compared with the spectra of the PVC and dicoumarol.

**Figure 3 medicina-55-00421-f003:**
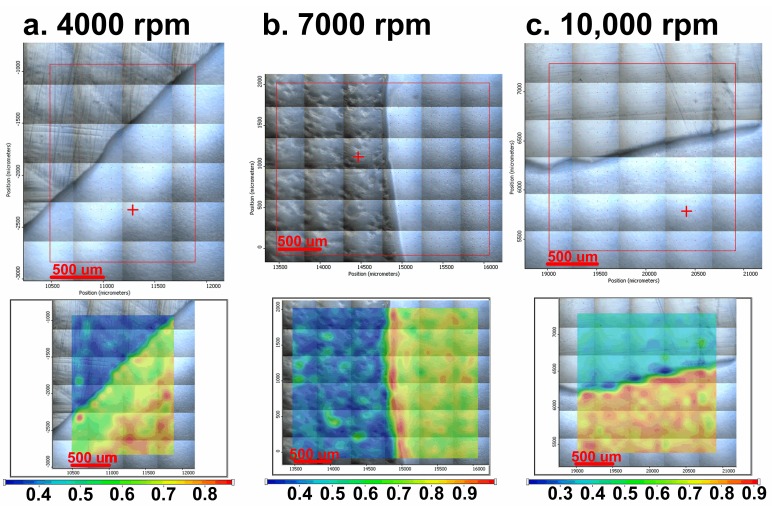
FT-IR microscopy images obtained at a wavelength of 1655 cm^−1^ for the deposition performed at 4000 rpm (**a**), at a wave length of 1650 cm^−1^ for the deposition performed at 7000 rpm (**b**), and at a wavelength of 1654 cm^−1^ for the deposition performed at 10,000 rpm (**c**).

**Figure 4 medicina-55-00421-f004:**
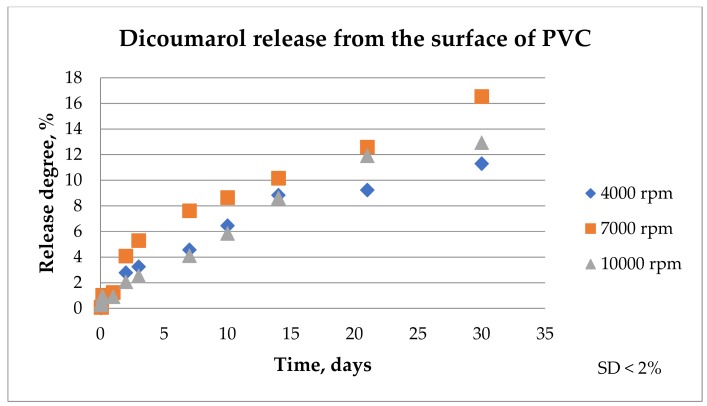
The Release of dicoumarol from the film obtained at 4000 rpm, 7000 rpm, and 10,000 rpm, respectively, analyzed by UV-VIS spectrometry at a wavelength of 303 nm standard deviation: SD < 2% in any of the represented points.

**Figure 5 medicina-55-00421-f005:**
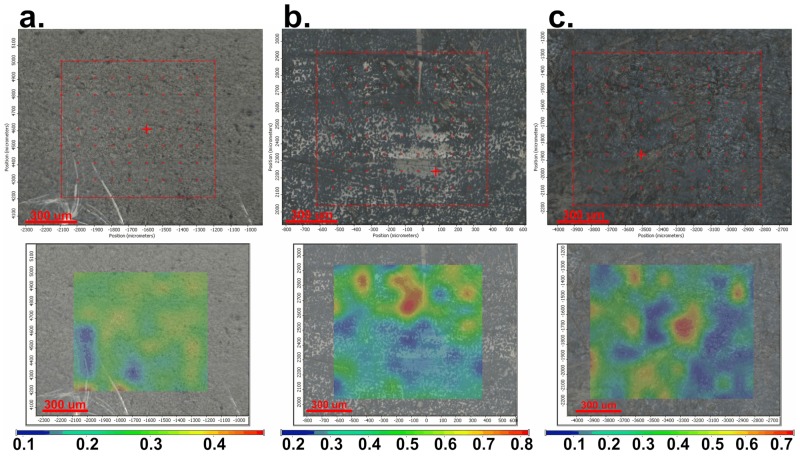
FT-IR images obtained at 1680 cm^−1^ for the film deposited at 4000 rpm: (**a**) after 7 days of immersion in simulated body fluid (SBF); (**b**) after 21 days of immersion in SBF; (**c**) after 30 days of immersion in SBF.

**Figure 6 medicina-55-00421-f006:**
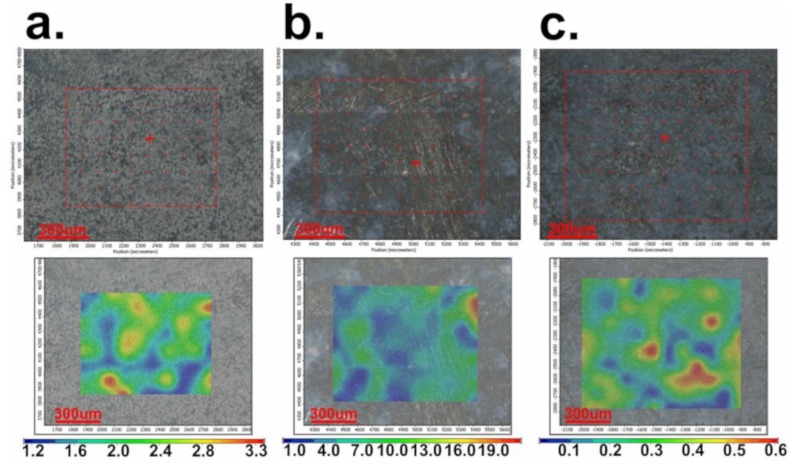
FT-IR images obtained at 1659 cm^−1^ for the film deposited at 7000 rpm: (**a**) after 7 days of immersion in SBF; (**b**) after 21 days of immersion in SBF; (**c**) after 30 days of immersion in SBF.

**Figure 7 medicina-55-00421-f007:**
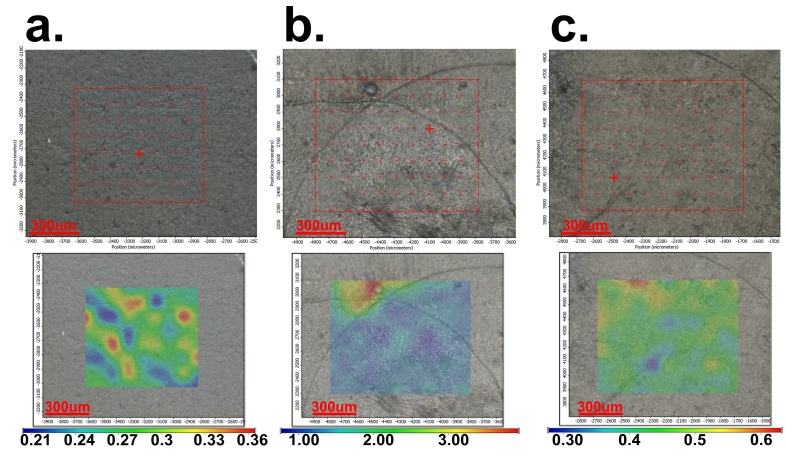
FT-IR images obtained at 1684 cm^−1^ for the film deposited at 10,000 rpm: (**a**) after 7 days of immersion in SBF; (**b**) after 21 days of immersion in SBF; (**c**) after 30 days of immersion in SBF.

**Figure 8 medicina-55-00421-f008:**
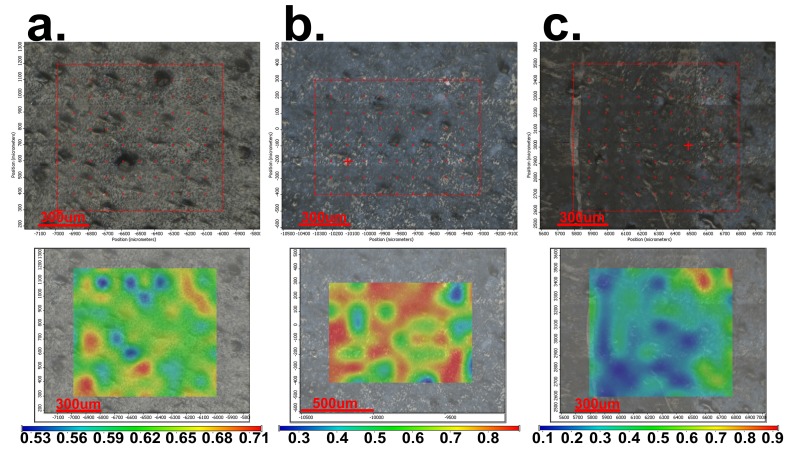
FT-IR images obtained at 1723 cm^−1^ for the uncovered PVC after: (**a**) 7 days; (**b**) 21 days; (**c**) and 30 days of immersion in SBF.

**Figure 9 medicina-55-00421-f009:**
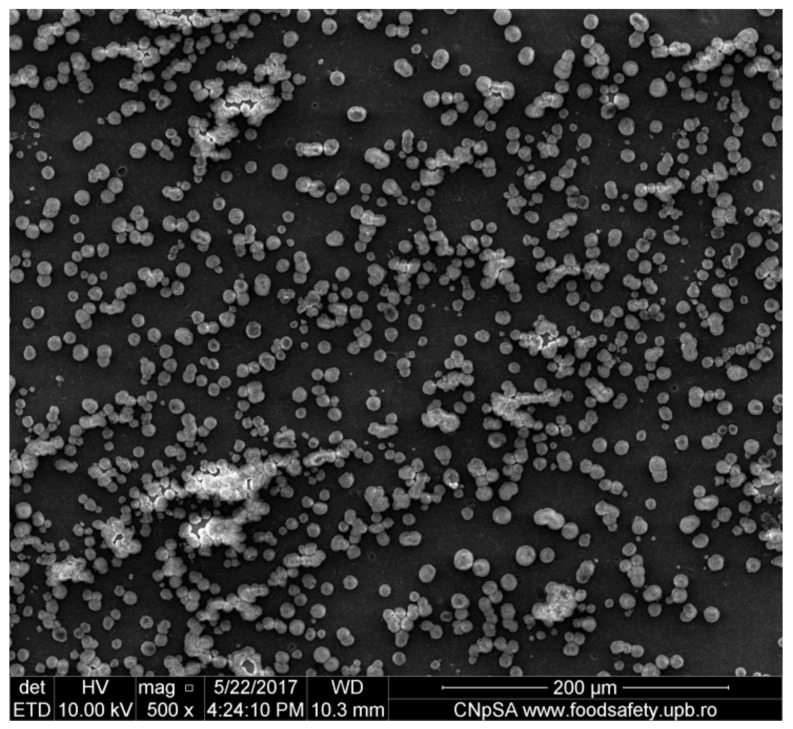
Scanning electron microscopy (SEM) image performed on the modified samples in order to have a better view of the salts deposited on the surface.

**Figure 10 medicina-55-00421-f010:**
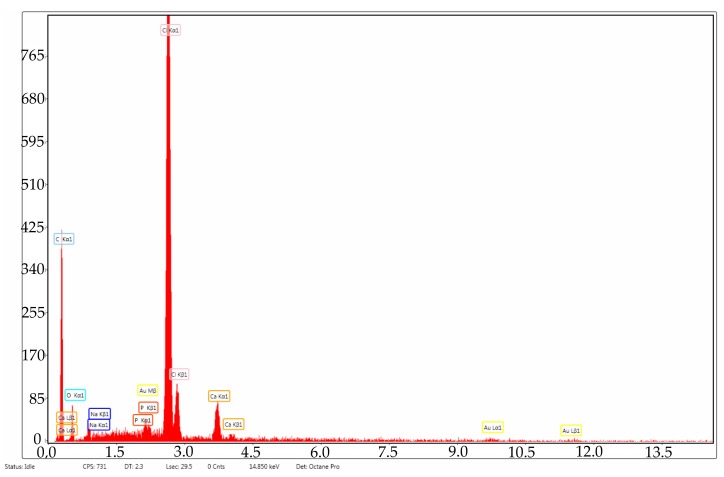
The Energy-dispersive X-ray spectroscopy (EDS) spectrum of the deposits present on the surface of the samples after 30 days of immersion in SBF.

**Figure 11 medicina-55-00421-f011:**
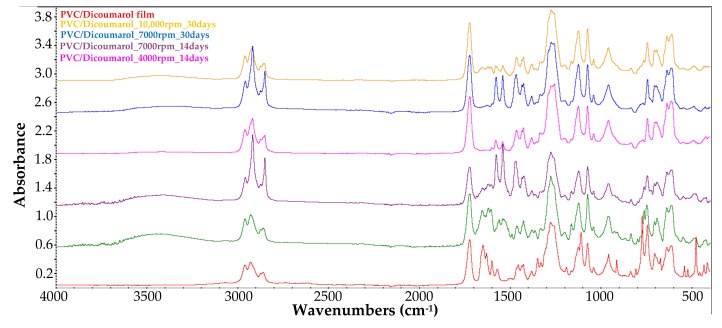
Compared FT-IR spectra of the polyvinyl chloride (PVC)/dicoumarol film before immersion in SBF, and after 14 and 30 days of immersion in SBF for the samples obtained at 4000 rpm, 7000 rpm, and 10,000 rpm.

**Figure 12 medicina-55-00421-f012:**
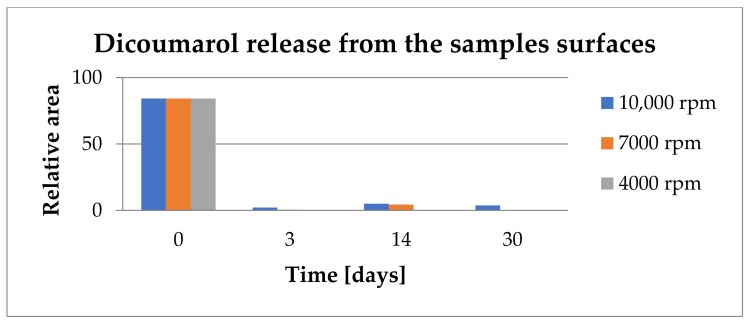
The changes that occur on the relative area of dicoumarol incorporated on the surface of the sample, in time, as the release from the polymeric film takes place.

**Figure 13 medicina-55-00421-f013:**
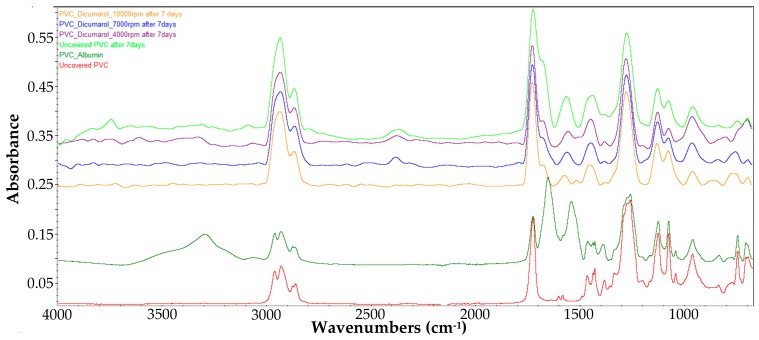
Compared FT-IR spectra of the PVC/dicoumarol films after 7 days of immersion in albumin solution of SBF, the spectra of the uncovered PVC before immersion, and the sample obtained by the drop cast method.

**Figure 14 medicina-55-00421-f014:**
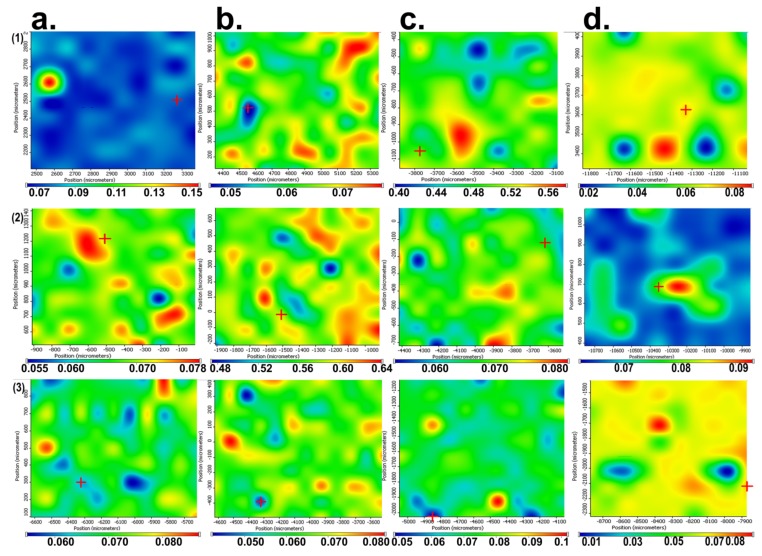
The Contour maps obtained by performing FT-IR microscopy on the samples obtained at 4000 rpm (**a**), 7000 rpm (**b**), 10,000 rpm (**c**), and on the uncovered PVC (**d**) after immersion in albumin solution of SBF after 7 days (**1**), 21 days (**2**), and 30 days (**3**).

**Figure 15 medicina-55-00421-f015:**
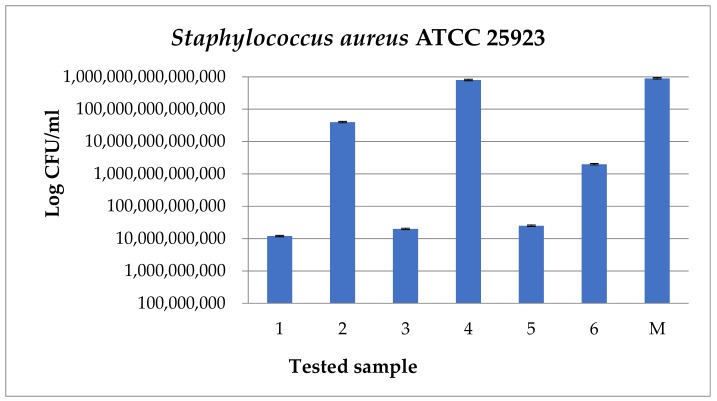
Graphic representation of the CFU/mL values for *Staphylococcus aureus* ATCC 25923 strain (ATCC—American Type Culture Collection).

**Figure 16 medicina-55-00421-f016:**
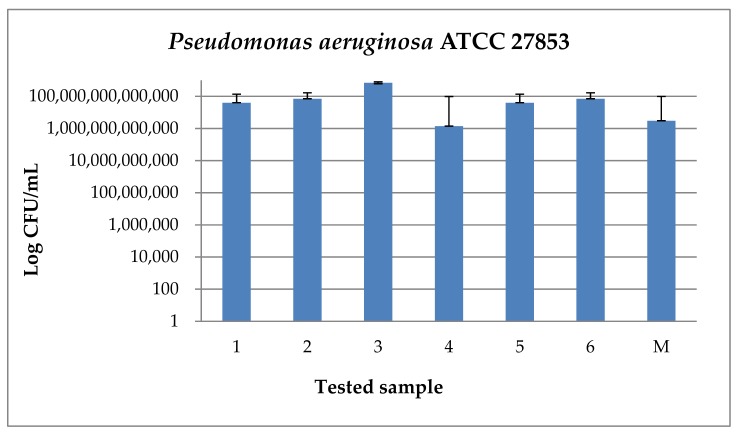
Graphic representation of CFU/mL values for *Pseudomonas aeruginosa* ATCC 27853 strain (ATCC—American Type Culture Collection).

**Table 1 medicina-55-00421-t001:** Work parameters for spin coating technique.

Nr. Crt	Dispencer	Spread	EBR	Dry
11	Spin acceleration: 1000 rpm	Spin acceleration: 100 rpm	Spin acceleration: 1000 rpm	Spin acceleration: 1000 rpm
	Spin speed: 100 rpm	Spin speed: 2000 rpm	Spin speed: 500 rpm	Spin speed: 4000 rpm
	Spin time: 5 ss, s	Spin time: 20 ss, s	Spin time: 10 ss, s	Spin time: 20 ss, s
22	Spin acceleration: 1000 rpm	Spin acceleration: 100 rpm	Spin acceleration: 1000 rpm	Spin acceleration: 1000 rpm
	Spin speed: 100 rpm	Spin speed: 2000 rpm	Spin speed: 500 rpm	Spin speed: 7000 rpm
	Spin time: 5 ss, s	Spin time: 20 ss, s	Spin time: 10 ss, s	Spin time: 20 ss, s
33	Spin acceleration: 1000 rpm	Spin acceleration: 100 rpm	Spin acceleration: 1000 rpm	Spin acceleration: 1000 rpm
	Spin speed: 100 rpm	Spin speed: 2000 rpm	Spin speed: 500 rpm	Spin speed: 10,000 rpm
	Spin time: 5 ss, s	Spin time: 20 ss, s	Spin time: 10 ss, s	Spin time: 20 ss, s

**Table 2 medicina-55-00421-t002:** Sample code.

Sample	Number
PVC/dicoumarol 4000 rpm	1
Albumin adsorbed PVC/dicoumarol 4000 rpm	2
PVC/dicoumarol 7000 rpm	3
Albumin adsorbed PVC/dicoumarol 7000 rpm	4
PVC/dicoumarol 10,000 rpm	5
Albumin adsorbed PVC/dicoumarol 10,000 rpm	6
PVC	M

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
