# Peer review of "Advanced Drug-Eluting Poly (Vinyl Chloride) Surfaces Deposited by Spin Coating"

_medicina, 2019, doi:10.3390/medicina55080421_

Round 1

Reviewer 1 Report

The manuscript “Advanced drug-eluting poly (vinyl chloride) surfaces deposited by spin coating” by Andronescu and coworkers reports an emerge method to modify the surface of polyvinyl chloride (PVC) by incorporating the active substance, dicoumarol.

Although the manuscript report a faceable approach to modify surfaces of medical devices, there are several comments that need to be addressed before publication in Medicina.

-         Manuscript need to be revised intensively for English language, structure and scientific concepts. for example:

·       Line 118: the amount “0,1”written using “,” instead of “.”

·       Line 138: “carbonic anhidrase” should be “carbonic anhydrase”

·       Line 194: “confirming his presence” using “his” is a clear mistake

·       Several places need specific punctuation, pronouns, and/or articles

-         Figures need more analysis and discussion. For example, Figure 2 describes FT-IR spectra at different stages of surface modifications. However, no clear discussion to the presented data. Line 193: “In the spectra …  the characteristic peaks of dicoumarol” There is no discussion or these peaks and at which wavenumbers are they ….

-         Line 223: “ FT-IR … in order to determine morphology”   It is not clear how FT-IR microscopy determine the “morphology and behaviour“ as reported. It is known that it can be used to determine the composition but morphology which could be determined using AFM.

-         Line 2.17: section “3.1. Release of dicoumarol from the thin film deposited on the surface of PVC”. It is not clear if the graph for release or recovery. The axis label is confusing….

Finally: major revision should be done though out the manuscript including structure and scientific discussion

Author Response

Response to the reviewer’s comment

Dear Editor,

Dear Reviewers,

First of all please let us thank you for your great effort to improve our manuscript, your comments being very much appreciated. We have attended to the comments and made the revision accordingly.

Reviewer 1:

1.   Manuscript need to be revised intensively for English language, structure and scientific concepts. for example:

·       Line 118: the amount “0,1”written using “,” instead of “.”

·       Line 138: “carbonic anhidrase” should be “carbonic anhydrase”

·       Line 194: “confirming his presence” using “his” is a clear mistake

·       Several places need specific punctuation, pronouns, and/or articles

Answer:

We appreciate the valuable comment from the reviewer. The manuscript was revised and the errors were corrected according to the comments received. Also, an addition language check was performed and most of the corrections are highlighted by track-change!

2.   Figures need more analysis and discussion. For example, Figure 2 describes FT-IR spectra at different stages of surface modifications. However, no clear discussion to the presented data. Line 193: “In the spectra …  the characteristic peaks of dicoumarol” There is no discussion or these peaks and at which wavenumbers are they ….

Answer:

The discussion regarding Figure 2 was modified and the missing information were added.

3. Line 223: “ FT-IR … in order to determine morphology” It is not clear how FT-IR microscopy determine the “morphology and behaviour“ as reported. It is known that it can be used to determine the composition but morphology which could be determined using AFM.

Answer:

The section was rewritten in order to be more clear. It is important to mention that, indeed, FTIR microscopy can offer similar results as AFM. For instance, it is possible to overlap the visual map with the FTIR map recorded at any wavelength, including the specific wavelengths of dicoumarol, PVC or deposits.

4.   Line 2.17: section “3.1. Release of dicoumarol from the thin film deposited on the surface of PVC”. It is not clear if the graph for release or recovery. The axis label is confusing….

Answer:

The axis label was replaced with „Release degree”

Thanking you again,

Best regards,

Oana Cristina Duta et al.

Reviewer 2 Report

Dear authors,

the manuscript provides reported to modify the surface of polyvinyl chloride (PVC) with a thin layer of dicouramol by spin coating process. The modified samples were characterized, analyzed and tested with bacteria. The functionalization and release of dicouramol on PVC improves the known properties of PVC.

Comments:

1 Introduction is not balanced. The authors have given too much space to the spin coating technique, which has been known for about 30 years. So, I suggest reducing this part. Instead, I advise the authors to increase the state of the art of antibacterial surfaces made from nanoparticle silver (Small 2018, 14, 1801219; Chem Rev. 2016 Mar 9;116(5):2826-85) to reinforce the use of dicouramol. 

2 To better clarify the validity of the data reported in figure 4, figure 12, figure 15 and figure 16 the data must have the statistical analysis.

3 The morphology characterization of bacteria on PVC with and without dicouramol would greatly enhance this work.

4 To facilitate the reading of the article I advise not to use sentences that are too long in the manuscript.

Author Response

Response to the reviewer’s comment

Dear Editor,

Dear Reviewers,

First of all please let us thank you for your great effort to improve our manuscript, your comments being very much appreciated. We have attended to the comments and made the revision accordingly.

Reviewer 2:

1. Introduction is not balanced. The authors have given too much space to the spin coating technique, which has been known for about 30 years. So, I suggest reducing this part. Instead, I advise the authors to increase the state of the art of antibacterial surfaces made from nanoparticle silver (Small 2018, 14, 1801219; Chem Rev. 2016 Mar 9;116(5):2826-85) to reinforce the use of dicouramol.

Answer:

We appreciate the valuable comments from the reviewer. The introduction was modified by reducing the spin coating technique description and inserting information about the background in the field of antimicrobial coatings.

2.    To better clarify the validity of the data reported in figure 4, figure 12, figure 15 and figure 16 the data must have the statistical analysis.

Answer:

      The standard deviation value was mentioned on the chart area of Figure 4 while for the figures 15 and 16 these values are presented directly on the figure – the space allowing the inclusion of these data directly on the figure.

3. The morphology characterization of bacteria on PVC with and without dicouramol would greatly enhance this work.

Answer:

At this moment no images were recorded. We will keep in our mind for the forthcoming works and we will prepare the requested protocols (immobilisation and drying of bacterial cells onto the samples; recording images, etc.).

4. To facilitate the reading of the article I advise not to use sentences that are too long in the manuscript.

Answer:

The manuscript has been revised and we tried to reduce long sentences in order to facilitate the reading of the article.

Thanking you again,

Best regards,

Oana Cristina Duta et al.

Round 2

Reviewer 1 Report

The manuscript is enhanced significantly for publication a Medicina. However, it still need graphic editing improvements.

Figures’ labels and names are not consistent in both font style and size. They need to be consistent and edited before publication.

Author Response

Dear Editor,

Dear Reviewers,

Thank you for your valuable comments. Your observations have led to the improvement of the manuscript.

Reviewer 1:

The manuscript is enhanced significantly for publication a Medicina. However, it still need graphic editing improvements.

Figures’ labels and names are not consistent in both font style and size. They need to be consistent and edited before publication.

Answer:

Thank you for your valuable comments. Figures labels and axis values were modified in order to be similar in font style and size, but also to be more easier to read.

Best regards,

Oana Cristina Duta et al.

Reviewer 2 Report

I thank the authors for the answers and modifications to the manuscript. I haven't other comments.

Author Response

Dear Editor,

Dear Reviewer,

Thank you for your valuable comments. Your observations have led to the improvement of the manuscript.

Best regards,

Oana Cristina Duta et al.